

# Infection of army ant pupae by two new parasitoid mites (Mesostigmata: Uropodina)

Adrian Brückner[1], Hans Klompen[2], Andrew Iain Bruce[3], Rosli Hashim[4] and Christoph von Beeren[1]

[1] Ecological Networks, Department of Biology, Technische Universität Darmstadt, Darmstadt, Germany
[2] Department of Evolution, Ecology and Organismal Biology, Ohio State University, Columbus, OH, United States of America
[3] School of Biological Sciences, Monash University, Melbourne VIC, Australia
[4] Institute of Biological Sciences, Faculty of Science Building, University of Malaya, Kuala Lumpur, Malaysia

## ABSTRACT

A great variety of parasites and parasitoids exploit ant societies. Among them are the Mesostigmata mites, a particularly common and diverse group of ant-associated arthropods. While parasitism is ubiquitous in Mesostigmata, parasitoidism has only been described in the genus *Macrodinychus*. Yet information about the basic biology of most *Macrodinychus* species is lacking. Out of 24 formally described species, information about basic life-history traits is only available for three species. Here we formally describe two new *Macrodinychus* species, i.e. *Macrodinychus hilpertae* and *Macrodinychus derbyensis*. In both species, immature stages developed as ecto-parasitoids on ant pupae of the South-East Asian army ant *Leptogenys distinguenda*. By piercing the developing ant with their chelicera, the mites apparently suck ant hemolymph, ultimately killing host individuals. We compare infection rates among all studied *Macrodinychus* species and discuss possible host countermeasures against parasitoidism. The cryptic lifestyle of living inside ant nests has certainly hampered the scientific discovery of *Macrodinychus* mites and we expect that many more macrodinychid species await scientific discovery and description.

## BACKGROUND

In 1982 David H. Kistner published an influential book chapter with the title ''The Social Insects' Bestiary'' (*Kistner, 1982*), a metaphor referring to the many thousand arthropod species exploiting social insect societies (*Kistner, 1979*; *Kistner, 1982*; *Hölldobler & Wilson, 1990*). Among them are such diverse groups as beetles, flies, wasps, ants, millipedes, silverfish, and mites (*Donisthorpe, 1927*; *Rettenmeyer, 1961*; *Kistner, 1979*; *Kistner, 1982*; *Hölldobler & Wilson, 1990*; *Buschinger, 2009*; *Parker, 2016*). The latter are particularly abundant guests of social insect colonies (*Kistner, 1982*; *Eickwort, 1990*; *Gotwald Jr, 1996*). The mite order Mesostigmata is notable in this respect because 20 out of 109 of its families are considered to have some kind of relationship with ants (*Walter & Proctor, 1999*;

Corresponding author
Adrian Brückner,
adrian.brueckner@gmail.com

*Beaulieu et al., 2011*). While most of the myrmecophilous mites use ant workers solely as transportation vehicles, some species are ectoparasitic (*Kistner, 1982*; *Eickwort, 1990*). For instance, *Macrocheles rettenmeyeri* (*Krantz, 1962*) (Mesostigmata: Macrochelidae) is an ectoparasite of Neotropical army ants (*Eickwort, 1990*). This 'myrmecophile' (ant lover) specifically attaches to the pulvilli of *Eciton dulcium* Forel, 1912 legs (*Krantz, 1962*; *Gotwald Jr, 1996*), where it probably sucks hemolymph from the ants' arolium, an adhesive organ at the tip of legs enabling ants to climb smooth or steep surfaces (*Hölldobler & Wilson, 1990*). While the negative impact of this ectoparasitic myrmecophile on host fitness is supposedly small, some of the ant-associated mites are parasitoids (*Lachaud, Klompen & Pérez-Lachaud, 2016*) and therefore, by definition (*Godfray, 1994*; *Godfray, 2004*), kill host individuals.

Given the great diversity of mite myrmecophiles, it is surprising that a parasitoid lifestyle is only known in a single mite family, i.e., the Macrodinychidae (Mesostigmata) (*González, Gómez & Mesa, 2004*; *Breton, Takaku & Tsuji, 2006*; *Krantz, Gómez & González, 2007*; *Lachaud, Klompen & Pérez-Lachaud, 2016*). In the most recent revisions of the group, the family's only genus, *Macrodinychus* Berlese, 1917, contained 24 valid species which are distributed throughout tropical regions and some temperate regions (*Kontschán, 2011*; *Kontschán, 2017*). Information about the basic biology and life history of most *Macrodinychus* species is lacking. The life cycle is only well known for three out of 24 species, i.e., *M. sellnicki* Hirschmann & Zirngiebl-Nicol, 1975 (*González, Gómez & Mesa, 2004*; *Krantz, Gómez & González, 2007*), *M. multispinosus* Sellnick, 1973 (*Lachaud, Klompen & Pérez-Lachaud, 2016*), and *Macrodinychus yonakuniensis* Hiramatsu, 1979 (*Breton, Takaku & Tsuji, 2006*). These species develop on ant pupae where immatures suck the host's hemolymph to an extent that is lethal to the ants (*Lachaud, Klompen & Pérez-Lachaud, 2016*). In 1975, Werner Hirschmann, a pioneer in the taxonomy of Uropodina, i.e., an infraorder within the order Mesostigmata (*Beaulieu et al., 2011*), hypothesized that all *Macrodinychus* species are parasites of ants (*Hirschmann, 1975*):

> "*Bei den Bodenfunden von* Macrodinychus *-Arten […] handelt es sich wohl um einzelne Zufallsfunde; denn der eigentliche Lebensraum der* Macrodinychus-*Arten scheint das Ameisennest zu sein, wo die Tiere als Paraphagen oder Parasiten an Ameisen leben dürften.*"

> *(Translation: The* Macrodinychus *species […] collected from soil samples are probably chance finds, because the actual living environment of the* Macrodinychus *species seems to be the ant nest, where the animals live as paraphages or parasites on ants.)*

When Hirschmann wrote these lines, his hypothesis was speculative and lacked solid evidence. For most *Macrodinychus* species we still lack information about their basic biology including possible symbiosis with ants. Today it is known that about one third of the *Macrodinychus* species are indeed associated with ants, with three definite examples of parasitoidism (*Lachaud, Klompen & Pérez-Lachaud, 2016*). In the present study, we provide further support for Hirschmann's hypothesis by adding two additional species to the list of *Macrodinychus* parasitoids. We formally describe and provide life history information

for two hitherto undescribed *Macrodinychus* species, *Macrodinychus hilpertae* Brückner, Klompen & von Beeren sp. nov. and *Macrodinychus derbyensis* Brückner, Klompen & von Beeren sp. nov. Both species were collected from colonies of the South-East Asian army ant *Leptogenys distinguenda*. Like other *Macrodinychus* parasitoids, the entire juvenile development of the new species took place as ecto-parasitoids on host pupae, ultimately killing the host individuals.

## MATERIALS & METHODS

### Collection and specimen depository

Two *Macrodinychus* (Mesostigmata: Uropodina) species were discovered during a project aiming to uncover the interactions of the army ant *Leptogenys distinguenda* and its diverse myrmecophile fauna (*Witte et al., 1999*; *Witte et al., 2002*; *Kistner, Witte & Maschwitz, 2003*; *Witte et al., 2008*; *Maruyama, von Beeren & Hashim, 2010*; *Maruyama, von Beeren & Witte, 2010*; *Mendes, von Beeren & Witte, 2011*; *Ott et al., 2015*). The mites were initially hidden, enclosed in ant pupal cocoons, and collection took place incidentally by collecting ant pupae (Fig. 1). The latter were collected during army ant colony emigrations using aspirators and forceps (for more information see *von Beeren et al., 2011b*). Collection took place in Malaysia, primarily at the Ulu Gombak Field Studies Centre of the University Malaya (latitude: 3.325, longitude: 101.750, elevation: 260) and additionally at the Biodiversity Institute Bukit Rengit (latitude: 3.596, longitude: 102.182, elevation: 72), between April and May 2008, August and September 2008, February and March 2009, August and September 2009, February and March 2010, and March and April 2011 (approx. 11 months in total). The specimens were stored in absolute ethanol and deposited at the Ohio State University Acarology Collection, Columbus, Ohio, USA (OSAL). Macrodinychid mites are vouchered together with their respective ant pupa. Further specimens are deposited at the Adam Mickiewicz University in Poznań (three specimens of *M. hilpertae* labeled as "Tank mite" and two specimens of *M. derbyensis* labeled as "Smooth shell"). Borrowing the latter specimens for morphological analysis was not possible in a reasonable time frame, because of an entire re-organisation of the department's mite collection. All other specimens have been lost during several institutional moves of one of the authors (CvB).

Note that the host ant is an undescribed species (K Arimoto, pers. comm., 2017). It was designated previously as *Leptogenys* sp. 1 (*Maschwitz et al., 1989*) and as *L. distinguenda* (see *Maschwitz & Steghaus-Kovac, 1991*; *Witte & Maschwitz, 2000*; *Witte & Maschwitz, 2002*; *Witte et al., 2008*; *von Beeren et al., 2011a*). To be consistent with the most recent publications we use the name *Leptogenys distinguenda* for the species, which is in fact a *nomen nudum*. Specimen images of *L. distinguenda* have been published previously (denoted there as *L. distinguenda* (*Maruyama, von Beeren & Hashim, 2010*)). Voucher ant specimens are deposited at the Southwest Forestry University Ant Collection, Kunming, Yunnan Province, China (collection identifiers: A11-5936–A11-5942).

The electronic version of this article in Portable Document Format (PDF) will represent a published work according to the International Commission on Zoological Nomenclature

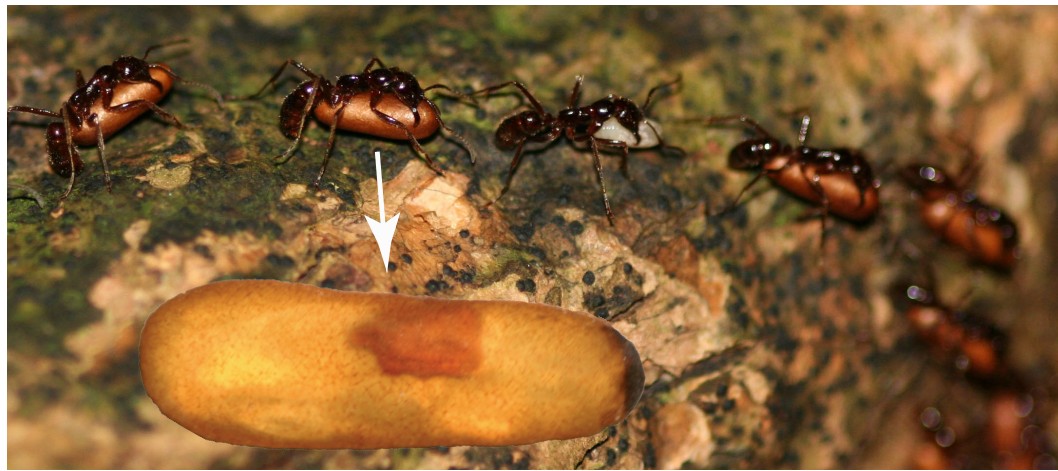

**Figure 1  Host pupa infected with *Macrodinychus* parasitoid.** Pupae were collected during colony emigrations of *Leptogenys distinguenda*. Pupal cocoons are opaque but become transparent in ethanol (white arrow). The highlighted pupa is infected with *Macrodinychus hilpertae*.

(ICZN), and hence the new names contained in the electronic version are effectively published under that Code from the electronic edition alone. This published work and the nomenclatural acts it contains have been registered in ZooBank, the online registration system for the ICZN. The ZooBank LSIDs (Life Science Identifiers) can be resolved and the associated information viewed through any standard web browser by appending the LSID to the prefix http://zoobank.org/. The LSID for this publication is: urn:lsid:zoobank.org:pub:84ADDB13-56F3-431D-9244-E19C3A2F7E04. The online version of this work is archived and available from the following digital repositories: PeerJ, PubMed Central and CLOCKSS.

## Prevalence of mites

To evaluate mite prevalence and screen for different development stages we gently opened a total of 2,360 *L. distinguenda* pupal cocoons from a total of six different colonies. Since many adult specimens found during these dissections have been lost, we were not able to reliably identify all *Macrodinychus* specimens to the species level. As a consequence of this, we could not determine the prevalence for each of the two species separately, but instead evaluated the overall parasitism rate among *Macrodinychus* spp., i.e., the total number of pupal infections by *Macrodinychus* mites. In addition, after an initial screening of 1,391 pupal cocoons for adult *Macrodinychus* mites in 2009 and 2010, all *L. distinguenda* pupae from different colonies were combined in 2012 for storage at the LMU Munich. Therefore, we only have limited data about the colony of origin of *Macrodinychus* mites.

For three additional *L. distinguenda* colonies we estimated the total number of pupae allowing us to estimate the number of pupal infections per colony. For this, *Leptogenys* bivouac sites were marked with tape and checked every 30 min for ongoing colony emigrations. Upon the start of an emigration, defined as workers carrying larvae or pupae to the new nest site, the number of *Leptogenys* workers carrying pupae and heading toward

the new bivouac site was repeatedly counted for 30 s, followed by a 150 s break till the emigration was finished. We did not collect pupae for dissections from these colonies.

## Morphological protocol and imaging

Specimens were dissected and slide mounted in Hoyer's medium or lactophenol (*Walter & Krantz, 2009*) and studied with bright-field, differential interference contrast and phase contrast microcopy. Morphological structures were drawn based on images taken during the phase contrast microscopy. In addition, focus-stacked images were taken with a Keyence VHX-5000 digital microscope (Keyence Deutschland GmbH, Neu-Isenburg, Germany) using the VH-Z50L lens. All measurements were taken using internal scale function as implemented in the Keyence system software (version 1.5.1.1; system version 1.0.4). A total of 37 images were uploaded to the Barcode of Life Database. Images can be accessed using the sample ID (provided in results) as search term. Images of all immature stages can be found on BOLD (search using the sample ID). Holotype label information is listed verbatim, with the different labels separated by forward slashes.

## Observations in laboratory nests

Interactions between host ants and adult *Macrodinychus* specimens were studied in laboratory nests containing 110–170 ant workers, 44–55 ant pupae, 22–30 callows (freshly hatched workers) and three to six clusters of ant larvae. Behavioral tests were carried out with workers of the myrmecophile's colony of origin. Details about the nest set-ups were described previously (*von Beeren et al., 2011a*). Myrmecophiles were tested individually. Frequently, myrmecophiles behaved excitedly for a short period after transferring them to laboratory nests, which sometimes initiated ant aggression. To avoid biases caused by the specimen transfer we gave myrmecophiles two minutes settling time before recording the ant behaviors. We then observed the interactions of the myrmecophile in the first 50 consecutive encounters with host ant workers (for definition of behavioral categories see Table S1). At the study time, we did not realize that there are two different *Macrodinychus* species and therefore the data presented here cannot be assigned to the species level. Nonetheless, we consider the behavioral data as valuable because behavioral interactions with host ants have not been studied systematically for any *Macrodinychus* species. To compare the host-symbiont interactions of *Macrodinychus* spp. with those of other *L. distinguenda* myrmecophiles, we additionally tested the following associates: the silverfish *Malayatelura ponerophila* (*Mendes, von Beeren & Witte 2011*) the spider *Sicariomorpha maschwitzi* (*Wunderlich, 1994*), the snail *Allopeas myrmecophilos* (*Janssen & Witte, 2002*), and the rove beetles *Maschwitzia ulrichi* (*Kistner, 1989*), *Witteia dentrilabrum* (*Maruyama, von Beeren & Hashim, 2010*), and *Togpelenys gigantea* (*Kistner, 1989*). Data on rove beetles were published previously (*von Beeren et al., 2011a*).

## Data analysis

Behavioral counts were expressed as compositional data (%) by standardizing for the total number of interactions per specimen (approx. 50 interactions per specimen: mean $\pm$ SD = 50.83 $\pm$ 3.20 interactions, $N = 97$). These multivariate data were analyzed with a permutational analysis of variance (PERMANOVA) with 9,999 permutations based

on Bray-Curtis similarities. Due to the rareness of certain associates, some specimens were tested multiple times (Table S2). This was considered in the PERMANOVA design (Myrmecophile species = fixed factor; Specimen ID = random factor). In addition to the multivariate analysis of behavioral interactions, we calculated an aggression index (AI in (%)) to measure the total aggression of ants towards a focal myrmecophile. For this, the sum of aggressive behaviors (chased, snapped, stung) was divided through the total number of interactions. We applied PERMANOVA for univariate cases based on Euclidean distances with the same design as described above. PERMANOVAs were run with the software Primer 7 (Primer-E Ltd., Ivybridge, UK, vers. 7.0.12) with the add-on PERMANOVA+1 (*Anderson, Gorley & Clarke, 2008*).

# RESULTS—TAXONOMIC SECTION

## Species descriptions

Infraorder **UROPODINA** Kramer, 1881
Family **MACRODINYCHIDAE** *Kontschán, 2017*
Genus **MACRODINYCHUS** *Berlese, 1917*

***Systematic note:*** For this study, we follow the classification of *Kontschán (2011)* and *Kontschán (2017)* in recognizing a single genus, *Macrodinychus Berlese, 1916*, in the family Macrodinychidae *Kontschán, 2017*. Within this genus four subgenera are recognized (largely corresponding to the "Stadiengattungen" of *Hirschmann, 1979*): *Macrodinychus, Monomacrodinychus Hirschmann, 1975* (= *Baloghmacrodinychus Hirschmann, 1979*, see *Halliday, 2015*), *Bregetovamacrodinychus Hirschmann, 1979*, and *Loksamacrodinychus Hirschmann, 1979*. Both of the new species belong in the subgenus *Macrodinychus* (*Monomacrodinychus*) based on the shape of the peritremes.

***Diagnosis of the genus Macrodinychus (based on Kontschán, 2011, Kontschán, 2017 and Hirschmann, 1975)***

Within Uropodina, the genus *Macrodinychus* is characterized by the following characters: Idiosoma large, oval or sometimes oblong, posterior margin rounded, anterior margin sometimes angular. Color yellow-brown to darkish brown. All legs short, but well developed. Tritosternum trifurcate with narrow basis. Gnathosoma with long hypostomal setae, horn-/peanut-like corniculi, pilose internal malae, pilose gnathotectum and chelicera with sclerotized nodes and without processes on tip fixed digit. Gnathosoma usually largely covered by coxae I. Genital shield of females small relative to the body (when compared to other Uropodina) and comparable in size to that of the males. Females and males do not differ in the shape and structure of the peritremes. Potentially viviparous.

### *Macrodinychus (Monomacrodinychus) hilpertae* Brückner, Klompen & von Beeren sp. nov.

*Type-host:* *Leptogenys distinguenda* (Formicidae: Ponerinae)

*Type-locality:* Ulu Gombak Field Studies Centre of the University Malaya (03°19.479′N, 101°45.163′E, altitude 230 m), Selangor, Malaysia.

*Type-specimens:* **Holotype:** female, accession number OSAL 0119286 (Fig. 2), stored in absolute ethanol, field sample code: cvb757macro008, deposited at OSAL (URL: https://acarology.osu.edu/database).

   **Paratypes:** on type host and from type locality: female, OSAL 0100050, 95% ethanol, cvb800macro002; male, OSAL 0106708, slide, dissected; on type host but from MALAYSIA: Pahang, Bukit Rengit (3.596 N 102.180 E, 72 m), female, OSAL 0103942, slide; female, OSAL 0103943, slide; female, OSAL 0103944, slide. All paratypes deposited at OSAL.

**Other specimens**: all on type host from type locality: female, OSAL 0119279, 95% ethanol, cvb757macro001; 5 deutonymphs, OSAL 0119280-281, 0119283-284, 0119290, 95% ethanol; 5 deutonymphs, OSAL 0102594, 0102596, 0106709-711, slide; 1 protonymph, OSAL 0119282, 95% ethanol.

*ZooBank registration:* Details of the new species have been submitted to ZooBank to comply with the current regulation of the ICZN. The Life Science Identifier (LSID) of the article is urn:lsid:zoobank.org:pub:84ADDB13-56F3-431D-9244-E19C3A2F7E04. The LSID for the new name *Macrodinychus* (*Monomacrodinychus*) *hilpertae* is urn:lsid:zoobank.org:act:88FADEC7-D4A5-4491-A45E-8F8176B65D31.

*Etymology:* Dedicated to Andrea Hilpert, for her advice, long lasting skillful technical assistance and support of AB and CvB.

**Description:** *General:* Length of the idiosoma 2,100 μm, width 1,250 μm (holotype). Shape oblong, posterior margin rounded, color darkish brown.

*Dorsal* (Fig. 2A): Dorsal shield rough with micro-ornamentation and an alveolar pattern. Completely sclerotized, middle part of the dorsal shield pronounced in a characteristic shape (also in lateral view). Dorsum hypertrichous. Dorsal shield covered by distinct and regularly distributed bulbiform setae. Setae covered with additional hairs on their margins. Dorsal and marginal shield not fused anteriorly. Tips of marginal shield not fused anteriorly. Marginal shield with a crenellation-like pattern of alveolae and ridges. Isolated pygidial shield with alveolar patterns, but without setae.

*Ventral* (Fig. 2C): Fused sternal and ventral shields bear an alveolar pattern with further micro-ornamentation on the rest of the cuticle. Female operculum between coxae II-III, length 188 μm, male operculum round, between coxae III, length 102 μm. Genital shields in both sexes without ornamentation. Scabellum covered by fish scale-like pattern. All ventral setae bulbiform. Position of sternal setae (*St*): *St 1* and *St 2* placed between coxae I and II. *St 3* inserted near the posterior margin of coxae II. Setation around the genital shield hypertrichous. An additional row of four pair of setae at the posterior margin of the operculum. Stigmata between coxae II and III, peritreme species-specific with finger-like branches (see Fig. S1C).

*Gnathosoma:* Gnathotectum triangular, extending in single peak with large barbs (length 190 μm). Salivary stylets (105 μm) long relative to gnathosoma. Subcapitulum: Corniculi

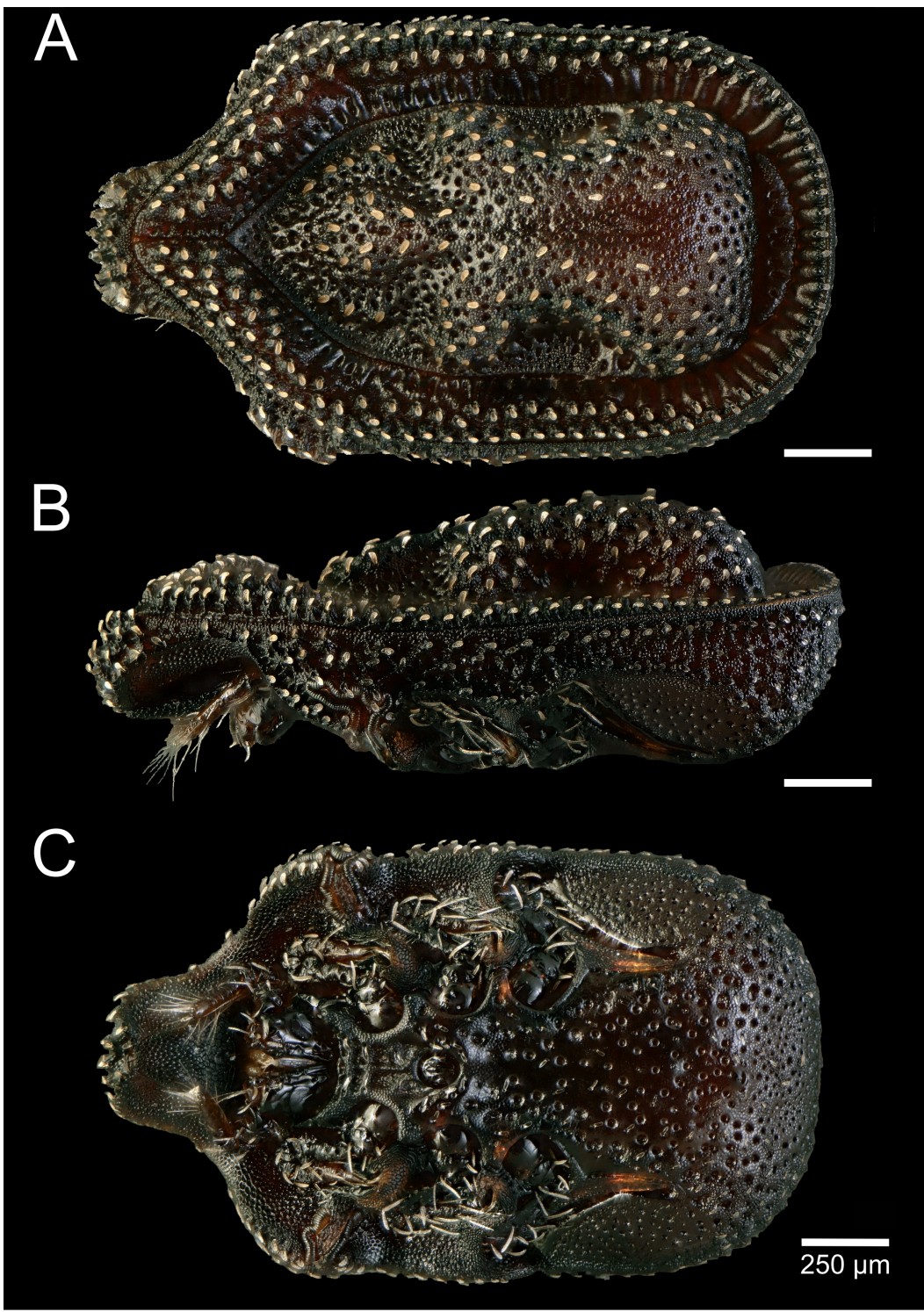

**Figure 2 _Macrodinychus_ (_Monomacrodinychus_) _hilpertae_ holotype.** (A) Dorsal, (B) lateral, and (C) ventral view of the _Macrodinychus hilpertae_ holotype. Scale bars are 250 μm.

peanut-shaped, blunt, length 40 μm, width 18 μm ($N = 2$). Hypostomal setae long (*h1* 59–62, *h2* 35–38, *h3* 41–55 μm), setiform, barbed; subcapitular (*sc*) setae rod-shaped, barbed (29–39 μm) (generally similar in arrangement and shape as in *M. derbyensis*, Fig. S1A, but setae shorter). Deutosternum poorly developed with 3 rows of 2 teeth each. Base of the tritosternum cylindrical, vase-like, with a smooth surface. Tritosternum trifurcate, laciniae with fine bristles (Fig. S1D). Chelicera with a distinct "nodus", and lacking a membranous extension on the fixed digit; distal end of fixed and movable digit with a small tooth, creating a bifid impression (Fig. S1E); moveable digit 70–76 μm, fixed digit 87 μm, entire fixed segment 280 μm (maximum width 41 μm), basal segment 232 μm ($N = 2$). Palp length 245 μm, width 42–49 μm ($N = 2$); tibia and tarsus fused, pretarsus in form of 4-tined apotele. Setation trochanter 2, femur 5, genu 4 setae (tibiotarsus not studied); trochanteral seta *v1* long, rod-shaped with increasing density of barbs towards tip (190 μm), *v2* setiform, substantially shorter, with much shorter barbs.

*Legs:* Legs short relative to body: leg I 1,050 μm, leg II 837 μm, leg III 859 μm, leg IV 950 μm. Note that the measurements provided here are better treated as approximations as the legs were folded into the pedofossae during measurements. Femora I–IV with small posterior flange. Leg setae long and barbed. Chaetotaxy (leg setation formula's following *Evans, 1963*, *Evans, 1972*): coxae 2-2-2-1; trochanters 4-5-5-5; femora 1 4/3 1, 1 4/2 1, 1 3/2 0, 1 3/2 0; genua 1 2/1 2/1 1, 1 2/1 2/1 1, 1 2/1 2/0 1, 1 2/1 2/0 1; tibiae 1 1/1 2/1 1, 1 1/1 2/1 1, 1 1/1 2/1 1, 1 1/1 2/1 1, tarsi II–IV 18 setae. Dorsal setae, especially on tibiae and genua, much shorter than lateral or ventral setae (measurements from single individual): on leg I tibia, respectively, 37 μm, 111 μm and 219 μm, leg II 45 μm, 75 μm, 123 μm; on genu leg I, respectively, 37 μm, 81 μm, 134 μm, leg II 50 μm, 74 μm, 93 μm. Setae *ad4* and *al4* on basitarsus IV long, well-developed, *pd4* and *pl4* much shorter and less barbed. Similar difference, though less pronounced, on other segments. Pretarsi of legs I small with a well-developed claw. Pretarsi of legs II –IV with long stalk, small pulvillus, and two claws.

**Immatures:** Larvae unknown. One protonymph, not studied. Deutonymphs weakly sclerotized. Dorsal cuticle distinctly ornamented, mid-dorsal setae distinctly shorter than marginal dorsal setae. Ventral setae generally short (30–40 μm), longer near the body margin. All sternal setae relatively short, *St4* also short relative to *St1-3* (26 μm vs. 70 μm, $N = 2$). Gnathosoma unclear in all available specimens. Chelicera with reduced, though distinct, fixed digit (∼1/2 as long as movable digit). Leg chaetotaxy as in adults except for femur I which carried 2 ventral setae (three ventral setae in adults); tibia IV lacking seta *pd1*. Legs I with small ambulacrum carrying claws.

**Differential diagnosis:** Within the genus *Macrodinychus*, this species can be distinguished from most others by its bulbiform setae. This character is only shared with *M. extremicus* (*Kontschán, 2011*), for which it was mistakenly identified in a previous publication (*Lachaud, Klompen & Pérez-Lachaud, 2016*). However, *M. hilpertae* can be unambiguously discriminated from *M. extremicus* by the following characters (for images of the *M. extremicus* holotype see Fig. S2): most prominently, species differ in the shape of their peritremes and the lateral shape of the dorsal shield (also clearly visible from the dorsal view as ring-like cavities). While *M. extremicus* has an undulating lateral shape of the dorsal shield with three mounds (Fig. S2), *M. hilpertae* possess just one mound without

any subdivision (compare Fig. 2A and Fig. S2A). In addition, *M. hilpertae* has a highly structured micro-ornamentation on the dorsal shield in contrast to *M. extremicus* (Figs. 2A and 2B). Alveolae on dorsal shield are bigger in *M. hilpertae*. Bulbiform setae are slenderer in *M. hilpertae* and distributed more evenly on the dorsal shield, while the setae are flap-like in *M. extremicus*, and are condensed at certain areas of the dorsal shield (Fig. S2), and are sometimes overlapping.

The nymphs of *M. hilpertae* differ substantially from those previously described from *M. (Macrodinychus) sellnicki* or *M. (Bregetovamacrodinychus) multispinosus*. In both of those species, nymphs have highly regressed idiosomal and leg setation, lacking nearly all idiosomal and non-tarsal leg setae. They also show highly reduced chelicera with fixed digits completely absent (*Krantz, Gómez & González, 2007*; H Klompen, pers. obs., 2017).

### *Macrodinychus (Monomacrodinychus) derbyensis* Brückner, Klompen & von Beeren sp. nov.

*Type-host:* *Leptogenys distinguenda* (Formicidae: Ponerinae)

*Type-locality:* Ulu Gombak Field Studies Centre of the University Malaya (03°19.479′N, 101°45.163′E, altitude 230 m), Selangor, Malaysia.

*Type-specimens:* **Holotype:** female, accession number OSAL 0119286 (Fig. 3), stored in absolute ethanol, field sample code: cvb757macro009, deposited at OSAL. **Paratypes:** On type host from type locality: male, OSAL 0106707, slide dissected; female, OSAL 0119292, cvb800macro001. **Other specimens**. On type host from type locality: three deutonymphs, OSAL 0119285, 0119288-289, 95% ethanol; four deutonymphs, OSAL 0102593, 0103953-955, slide; 1 protonymph, OSAL 0119291, 95% ethanol, cvb757macro013; 1 protonymph, OSAL 0102595, slide. All specimens deposited at OSAL.

*ZooBank registration:* Details of the new species have been submitted to ZooBank to comply with the current regulation of the ICZN. The Life Science Identifier (LSID) of the article is urn:lsid:zoobank.org:pub:84ADDB13-56F3-431D-9244-E19C3A2F7E04. The LSID for the new name *Macrodinychus* (*Monomacrodinychus*) *derbyensis* is urn:lsid:zoobank.org:act:83C74026-EF1F-41A8-915F-49B0B2188E2A.

*Etymology:* The name of the new species refers to the fact that both new species (*M. hilpertae* and *M. derbyensis*) co-occur in the same host species. We further picked this name to honor one of the most diversity-loving, integrative and awesome sports—roller derby—with all its players, flamingos, zebras and enthusiastic supporters.

*Description:* *General:* Length of the idiosoma 2,370 µm, width 1,480 µm (holotype). Shape oblong, anterior and posterior margins rounded, color ocher-brown/ brown.

*Dorsal* (Fig. 3A)*:* Dorsal shield completely sclerotized and smooth without ornamentation, but with deep alveolae in the middle part of the dorsal shield. Dorsal shield covered by distinct and regularly distributed smooth, needle-like setae. Setation hypertrichous. Dorsal and marginal shield not fused anteriorly. Tips of marginal shield not clearly distinct anteriorly and sub-marginal shield apically fused with marginal shield. Marginal shield with a crenellation-like pattern posterior, no pattern medial and an alveolar pattern anterior. Isolated pygidial shield with eight setae and deep pits.

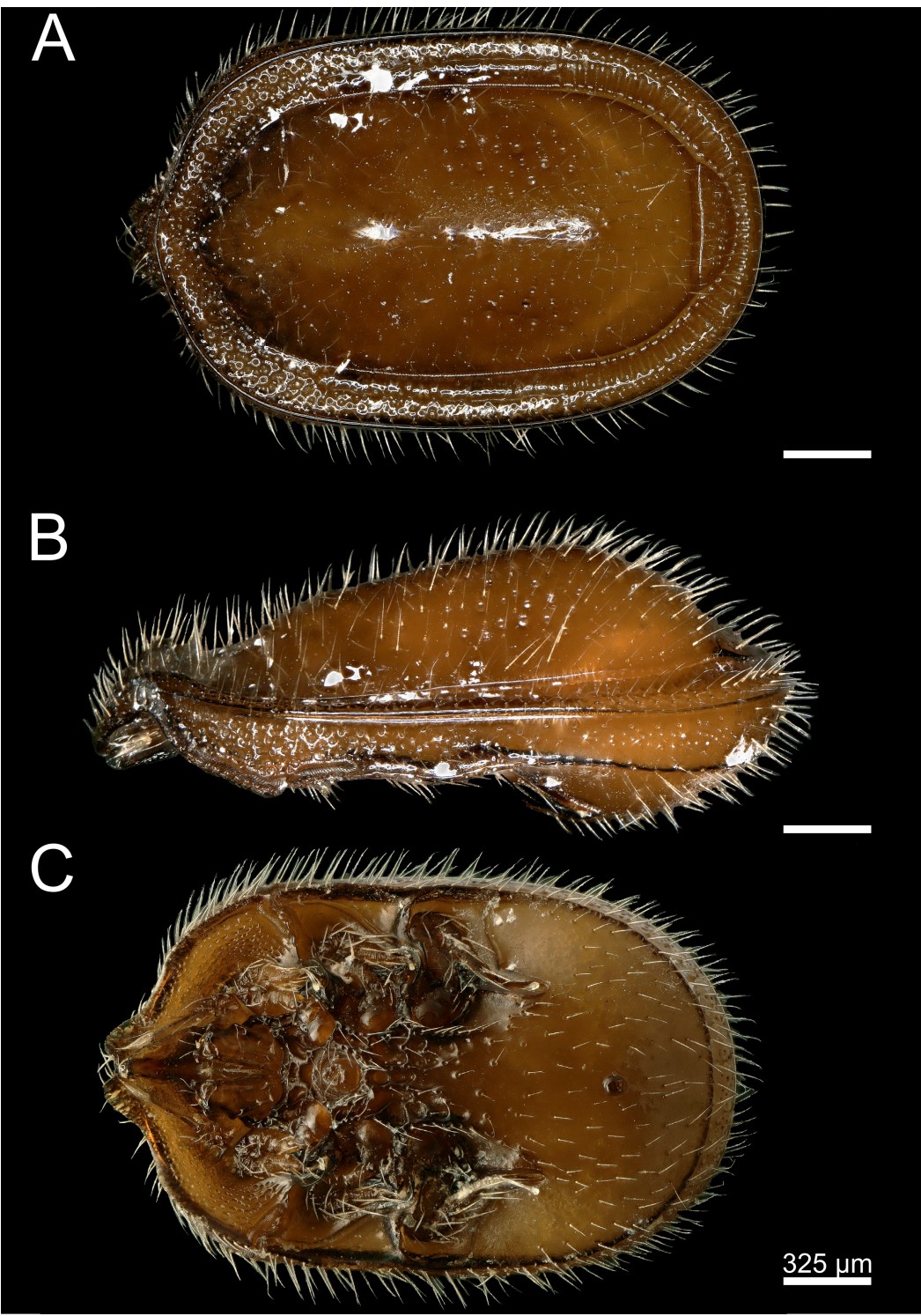

**Figure 3** *Macrodinychus* (*Monomacrodinychus*) *derbyensis* **holotype.** (A) Dorsal, (B) lateral, and (C) ventral view of the *Macrodinychus derbyensis* holotype. Scale bars are 325 μm.

*Ventral* (Fig. 3C)*:* Ventral shield posterior smooth without ornamentation and covered with setae, but ventral shield anteriorly covered with a fine structured micro-alveolar pattern without setae (starting at the peritremes). Sternal and ventral shield fused. Cuticle posterior to genital shield between coxae III and IV with deep pit-like ornamentation. Female operculum between coxae II-III, length 196 μm, male operculum round, between coxae III, length 140 μm. Genital shields in both sexes without ornamentation. Scabellum covered by a fine ornamentation. All ventral setae smooth and needle-like. Only two pairs of sternal setae (*St1* and *St2*) clearly distinct from a row of setae along the endopodal line, and rows of additional pairs of setae at the posterior margin of the operculum. Position of sternal setae (*St*): *St 1* placed between coxae I and II. *St 2* inserted near the anterior margin of coxae II. Stigmata between coxae II and III, peritreme species-specific with finger-like branches (Fig. 3B; Fig. S1B).

*Gnathosoma:* Gnathotectum triangular, extending in single peak with strong barbs (length 254 μm). Salivary stylets thick and long (236 μm). Subcapitulum (Fig. S1A): Corniculi peanut-shaped, blunt, length 61 μm, width 22 μm ($N = 2$). Hypostomal setae long (*h1* 101–108 μm, *h2* 58–61 μm, *h3* 91 μm), setiform, barbed, subcapitular setae shorter (46–48 μm), rod-shaped, barbed. Deutosternum poorly developed with three rows of two teeth each. Chelicera with "nodus", lacking membranous extensions on fixed digit; movable digit 82 μm, fixed digit 102 μm, entire fixed segment 390 μm (maximum width 46 μm), basal segment 390 μm. Palp length 310 μm, width 42–47 μm ($N = 2$); tibia and tarsus fused, pretarsus in form of 4-tined apotele. Setation trochanter 2, femur 5, genu 4 setae (tibiotarsus not studied); trochanteral seta *v1* rod-shaped with increasing density of long barbs towards tip (length 124 μm), *v2* setiform, substantially shorter.

*Legs:* Legs short relative to body: leg I 1,184 μm, leg II 1,119 μm, legs III–IV 1,130 μm ($N = 1$). Femora I–IV with small posterior flange. Setal shape and chaetotaxy as in *M. hilpertae.* Dorsal setae on tibiae and genua much shorter than lateral or ventral setae (measurements from single individual): on leg I tibia, respectively, 59 μm, 192 μm and 365 μm, leg II 45 μm, 75 μm, 123 μm; on genu leg I, respectively, 34 μm, 85 μm, 169 μm, leg II 77 μm, broken, 154 μm. Differentiation of setae *al4* and *ad4* vs. *pd4* and *pl4* on basitarsus IV distinct, but less so than in *M. hilpertae.* Pretarsi of legs I small with a single claw. Pretarsi of legs II–IV each with a long stalk, well-developed pulvillus including a pair of setiform structures, and two claws. Pulvillus distinctly larger than in *M. hilpertae.*

**Immatures:** Larvae unknown. Two protonymphs, not studied. Deutonymphs weakly sclerotized. Dorsal cuticle without distinct ornamentation, mid-dorsal setae as long as marginal dorsal setae. Ventral setae fewer than in *M. hilpertae*, but longer (80–95 μm). Sternal setae long *St4* about 2/3 the length of *St1-3* (81 μm vs. 128 μm, $N = 3$). Gnathosoma well developed compared to other American macrodinychid species. Chaetotaxy as in adults: *h1* 41 μm, *h2* 67 μm, *h3* 91 μm, *sc* 74 μm ($N = 1$), corniculi peanut–shaped, 28 × 15 μm. Chelicera shorter than in adult, fixed digit reduced, but distinct (∼1/2 length of movable digit). Palps weakly developed, palp apotele present, 3–4-tined. Leg chaetotaxy as in adults but femur I with only two, rather than three, ventral setae; ventral setae slightly longer than dorsal ones. Legs I with small ambulacrum carrying claws.

**Differential diagnosis:** Within the genus *Macrodinychus*, this species can be distinguished from most others by its isolated pygidial shield bearing smooth and needle-like setae. This character is only shared with *M. vietnamensis* (*Hirschmann, 1983*), which is the morphologically closest relative. The holotype of *M. vietnamensis* is lost and not deposited at the Natural History Museum in Budapest, Hungary, as stated in the formal description of the species (*Hirschmann, 1983*). However, we found a slide-mounted specimen designated by Hirschmann as *M. vietnamensis* at the Bavarian State Museum of Zoology. We used this specimen for comparisons (see Fig. S3). The following characters can be used to discriminate the species: the cuticle of *M. derbyensis* posterior to genital shield between coxae III and IV possess a deep pit-like ornamentation (Fig. 3C), a character which is absent in *M. vietnamensis*. Furthermore, the sub-marginal shield of *M. derbyensis* is apically fused with the marginal shield, while the sub-marginal shield is apically distinct from the dorsal shield and the marginal shield in *M. vietnamensis*. In addition, *M. derbyensis* has rows of setae along the endopodal line and additional rows of paired setae at the posterior margin of the operculum, while *M. vietnamensis* has only five pairs of sternal setae.

Descriptions of immature morphology of *Macrodinychus* are hitherto really limited. Comparing the two species described here, the deutonymphs of *M. derbyensis* differs from those of *M. hilpertae* in relative length of dorsal and ventral setae (especially relative lengths of *St4* vs. *St1-3*), lack of ornamentation of the dorsum, and presence (vs. absence) of seta *pd1* on tibia IV.

## Key to species of the subgenus *Monomacrodinychus* (updated from *Kontschán, 2011* and *Kontschán, 2017*)

To aid in differentiation of the new species from previously described species, we updated the key to species in the subgenus *Monomacrodinychus* (*Kontschán, 2011*; *Kontschán, 2017*):

**1.** Peritreme without branches ................................. other subgenera (see *Kontschán, 2011*)
Peritreme with finger-like branches ............................................................................ **2**
**2.** Isolated pygidial shield absent ................................................. *M. multipennus*
Isolated pygidial shield present ....................................................................... **3**
**3.** Dorsal and ventral shields with bulbiform setae ............................................. **4**
Dorsal and ventral shields without bulbiform setae ........................................... **5**
**4.** Dorsal shield with rough alveolae without microstructural reticular ornamentation, bulbiform seate big and flap-like, two half ring-form cavities in the central region of the dorsal shield ...................................................................... *M. extremicus*
Dorsal shield alveolae with microstructural reticular ornamentation, bulbiform setae smaller and distinct from each other, half ring-form cavities less pronounced ..................................................................................................... *M. hilpertae*
**5.** Dorsal setae smooth ..................................................................................... **6**
Dorsal setae with hairs on their margins ......................................................... **9**
**6** Isolated pygidial shield with setae .............................................................. **8**
Isolated pygidial shield without setae ............................................................ **7**
**7.** Pygidal shield narrow, anterior horns absent.......................................*M. kaszabi*
Pygdial shield hemispherical, anterior horn present............................................ *M. tanduk*

**8.** No rows of setae along endopodal line, five pairs of sternal setae, cuticle posterior to genital shield between coxae III and IV without ornamentation, sub-marginal shield apically distinct form dorsal shield and marginal shield ............................ *M. vietnamensis*
Rows of setae along the endopodal line, additional rows of paired setae at the posterior margin of the operculum, only *St1* and *St2* clearly distinguishable, cuticle posterior to genital shield between coxae III and IV with deep pit-like ornamentation, sub-marginal shield apically fused with marginal shield ........................................*M. derbyensis*
**9.** Isolated pygidial shield with setae ...................................................... *M. shibai*
Isolated pygidial shield without setae ................................................................... **10**
**10.** Apical part of dorsal setae wide and bear short hairs .................................. *M. yoshidai*
Dorsal setae needle-like with hairs on their margins .......................................**11**
**11** Alveolar ornamentation on the lateral part of the dorsal shield, genital shield of female with alveolar pattern ............................................................................ *M. baloghi*
Alveolar ornamentation on the whole dorsal shield, genital shield of female without pattern ............................................................................................................... *M. hirschmanni*

## RESULTS—LIFE HISTORY SECTION

### Host infection rate

Out of 2,360 inspected *L. distinguenda* pupae 40 were infected with one of the two *Macrodinychus* species, i.e., the pupal infection rate at Ulu Gombak was 1.69%. Each pupa was only infected by a single *Macrodinychus* specimen. The inspection of host pupae from a single colony in 2009 demonstrated that *Macrodinychus* species can co-occur in the same colony (*M. hilpertae* = 2 infected pupae; *M. derbyensis* = 4 infected pupae).

The pupal number per colony was estimated for three different *L. distinguenda* colonies: 6,456 pupae, 5,845 pupae, and 9,846 pupae. With an infection rate of 1.69%, the total number of pupal infection per colony was estimated to be 109, 99, and 166, respectively.

### Life-history of *M. hilpertae* and *M. derbyensis*

The dissection of 2,360 ant pupae recovered 20 immature and 20 adult mite stages. The following two immature development stages were found: three protonymphs (*M. derbyensis*, $N = 2$; *M. hilpertae*, $N = 1$) and 17 deutonymphs (*M. derbyensis*, $N = 7$; *M. hilpertae*, $N = 10$). All parasitized ant pupae had small, brownish scars (Fig. 4), which were not present in unparasitized ant pupae. Adult mites and deutonymphs that were detached from ant pupae left behind a conspicuous abnormal cavity in pupal bodies (Fig. 4). We did not find *Macrodinychus* larvae. However, we detected larval exuviae of three *M. hilpertae* individuals (Fig. 4). Exuviae of proto- and deutonymph were frequently detected inside pupal cocoons, often still attached to the mite or to the ant specimen (e.g., see BOLD image of sample 'cvb757macro002').

### Observations in laboratory nests

Myrmecophile species differed in their behavioral interactions with host ants (PERMANOVA, pseudo-$F = 37.27$, $P < 0.001$; Fig. 5A). *Macrodinychus* specimens generally walked slowly in the laboratory nests among host workers, which primarily

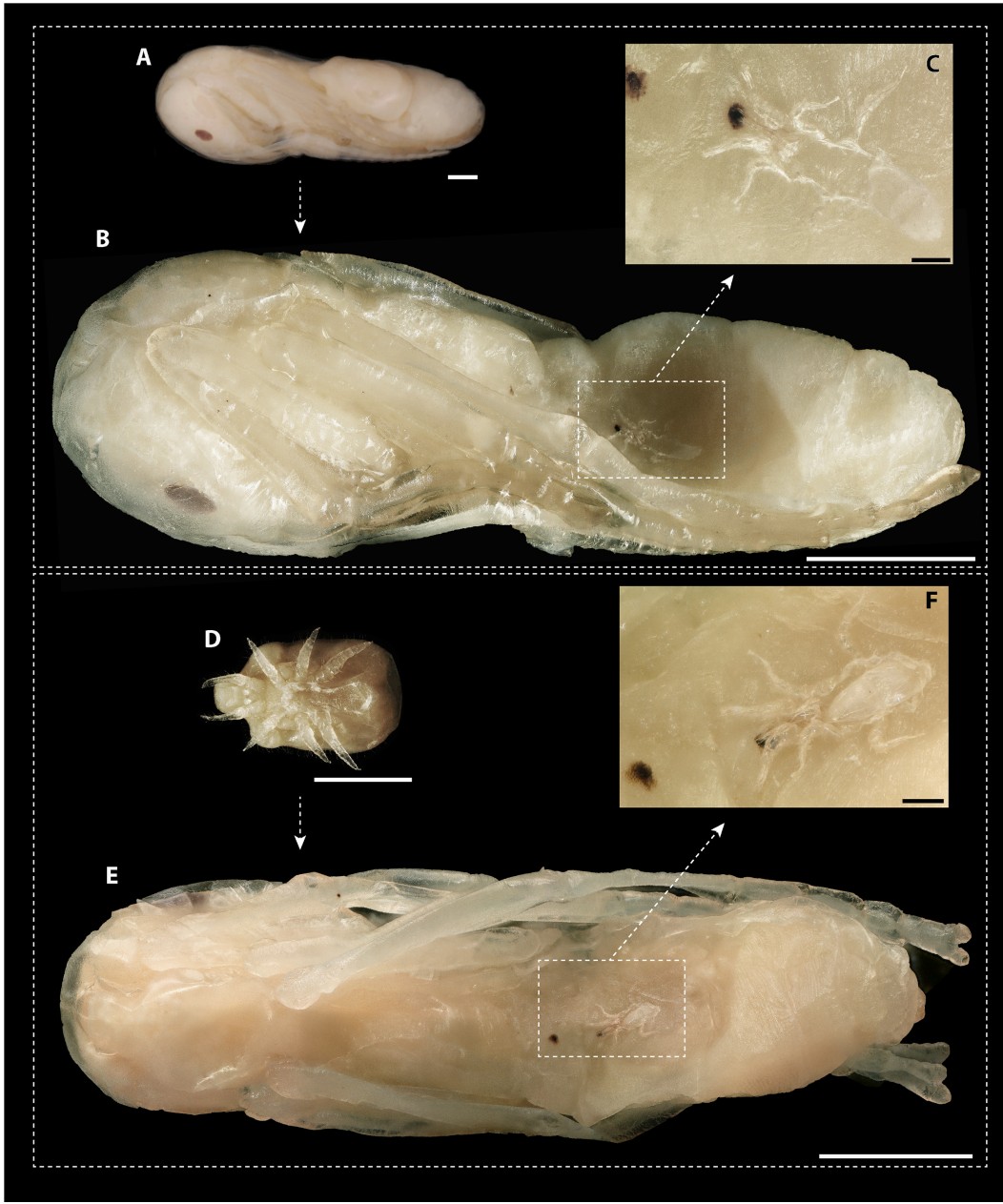

**Figure 4** ***Macrodinychus hilpertae* attached to ant pupae.** (A, D) Deutonymphs of *M. hilpertae* (A) attached to and (D) detached from developing ant pupae (silk cocoon removed). (B, E) Respective ant specimens with mites removed exposing the abnormal intrusion in the ants' gasters and the brownish scars. Larval exuviae of *M. hilpertae* are still sticking to the ants (dashed square). (C, F) Enlarged view of the larval exuviae. The cheliceral cuticles are still sticking to the ant's wound. Scale bars are 1 mm except for images c and f where it is 0.1 mm.

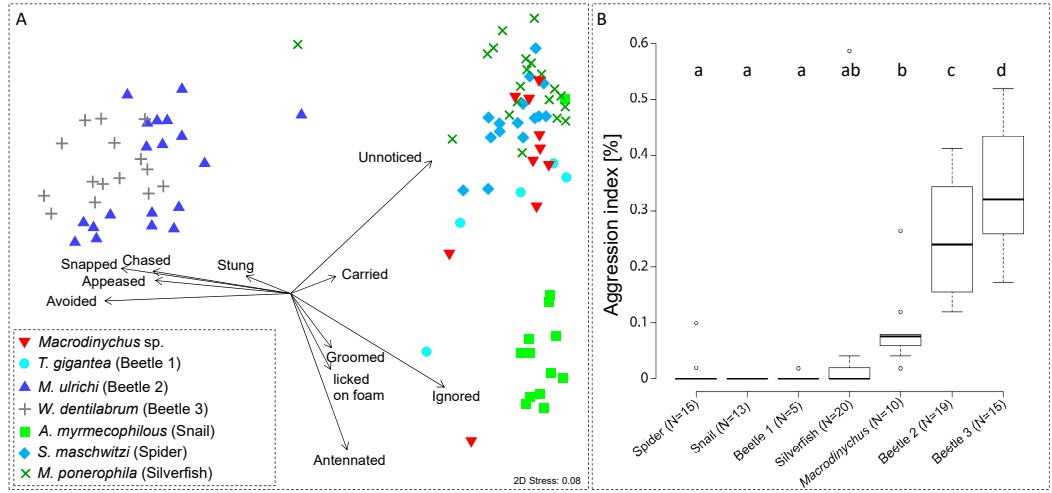

**Figure 5  Ant-symbiont interactions and ant aggression towards symbionts.** (A) Nonmetric-multidimensional scaling (NMDS) plot visualizes the differences in behavioral interactions between host ants and seven symbiont species. Each data point represents approx. 50 encounters of an individual symbiont with host ants. Length and direction of arrows visualize the contribution of behavioral categories to data separation. For clarity, the origin of arrows is not centered in the plot. "Stress" quality measure of the NMDS. (B) Aggressive behaviors of ants towards symbionts. Depicted is the aggression index (AI), which is the proportion of aggressive behaviors (sum of chased, snapped, and stung) towards symbiont specimens relative to their total number of interactions (approx. 50 for all specimens). Different letters depict significant differences ($p < 0.05$; PERMANOVA pairwise tests).

did not notice (mean $\pm$ SD: $24 \pm 10$ events, $N = 10$) or ignored the mites (mean $\pm$ SD: $18 \pm 8$ events, $N = 10$; Fig. 5). Out of ten behavioral tests, mites were picked up in nine cases by ant workers (see Video S1). This interaction was initiated by intense antennation by an approaching ant (mean $\pm$ SD: $4 \pm 4$ events, $N = 10$). Ants then carried around the mites (mean $\pm$ SD: $1 \pm 0$ events, $N = 10$; Table S2), an interaction uniquely found in *Macrodinychus* mites and in the snail *Allopeas myrmekophilos* (*Witte et al., 2002*). We observed that ants often dumped the mites at the ants' refuse site (see Video S1).

Ant aggression towards separate myrmecophile species differed (PERMANOVA, pseudo-$F = 15.00$, $P < 0.001$; Fig. 5B). Compared to other myrmecophiles of *L. distinguenda*, ants attacked *Macrodinychus* mites at a moderate level (mean (AI) $\pm$ SD: $0.08 \pm 0.07$, $N = 10$; Fig. 5B). During the ∼50 encounters with host ants, ants chased (mean $\pm$ SD: $0.10 \pm 0.32$ events, $N = 10$), snapped (mean $\pm$ SD: $2.40 \pm 1.78$ events, $N = 10$) and tried to sting the mites (mean $\pm$ SD: $2.0 \pm 2.05$ events, $N = 10$). All *Macrodinychus* specimens survived the ∼50 encounters with host ants.

# DISCUSSION

## Life-history of macrodinychid mites

Adopting Kistner's metaphor of a social insect bestiary, the two herein described *Macrodinychus* parasitoids are extraordinary examples of specialized beasts invading ant colonies. Both species fulfill their immature development inside army ant colonies,

which constitutes a stable and predator-free space with sufficient supply of food (*Kistner, 1979*; *Hölldobler & Wilson, 1990*; *Hughes, Pierce & Boomsma, 2008*). More specifically, *Macrodinychus* immatures were attached to and most likely fed on defenseless ant pupae.

While we do not provide direct evidence here that *M. hilpertae* and *M. derbyensis* were feeding on the host's hemolymph or tissue, this seems to be the most parsimonious explanation to us. First, parasitized pupae possessed scars which can be interpreted as feedings marks. We consider it most likely that scars represent areas where mites used their chelicera to pinch the ant's cuticle in order to feed on host tissue and/or drink from the excreting hemolymph. Second, we found exuviae of different development stages inside individual pupal cocoons, indicating that the mites grew by feeding on the ant pupae. Consumption of host tissue/hemolymph is also indicated by the fact that detached mites left behind physical impressions constituting substantial parts of the ants' gasters.

In ant-associated Mesostigmata, parasitoidism has only been described in the genus *Macrodinychus* (*Lachaud, Klompen & Pérez-Lachaud, 2016*). The five more extensively studied *Marodinychus* species (including *M. hilpertae* and *M. derbyensis*) seem to share the following key life-history traits (*González, Gómez & Mesa, 2004*; *Breton, Takaku & Tsuji, 2006*; *Krantz, Gómez & González, 2007*; *Lachaud, Klompen & Pérez-Lachaud, 2016*): all species seem to fulfill their entire immature development, including larval, proto- and deutonymphal stage, by feeding on individual ant pupa. For this, they seem to pierce the pupal cuticle with their chelicera to consume host tissue and/or to suck host hemolymph leaving behind small, brownish feeding marks. The larvae have well-developed legs and hence seem to be the mobile instar to find suitable hosts, while proto- and deutonymphs are more likely immobile feeding instars. *Macrodinychus* adults finally occupy a substantial part of the pupa's body. Once removed from the ant, they leave behind a conspicuous cavity, providing visual evidence for a lethal feeding strategy.

## Parasitism rates of macrodinychid mites—native versus invasive host ants

Besides similarities among species, we also detected a notable difference between the *Macrodinychus* species studied previously and those studied here. The prevalence of infection, measured as the percentage of infected to non-infected host pupae, was markedly lower in *M. hilpertae* and *M. derbyensis* (approx. 2% vs. 15%–90% in other *Macrodinychus* species; see *González, Gómez & Mesa, 2004*; *Breton, Takaku & Tsuji, 2006*; *Krantz, Gómez & González, 2007*; *Lachaud, Klompen & Pérez-Lachaud, 2016*). Various explanations could be responsible for the vastly different parasitism rates among studied macrodinychid mites. For example, the particular sampling methods or seasonal and spatial differences in parasitoid prevalence could conceivably cause such variation. Another possible cause is that *Macrodinychus hilpertae* and *M. derbyensis* have been studied in a native host-parasitoid system, while other *Macrodinychus* spp. have exclusively been studied in association with invasive ant species. Parasitoids are often a major source of host mortality and intense selection on the host to evolve counter-defenses against parasitoid attacks can be expected (*Godfray, 1994*). In species drifts across continents, however, local and alien interaction partners have no coevolutionary history (*Thompson, 2005*; *Simberloff, 2013*).

In such situations, naïve hosts can suffer from extremely high parasitism rates (*Kirk, 2003*; *Prenter et al., 2004*; *Lymbery et al., 2014*), which can, in extreme cases, lead to the decline of local host populations (*Holdich & Reeve, 1991*). Invasive ants, in particular, might be predisposed to parasitism by local species due to mass propagation coupled with genetic depletion (*Sakai et al., 2001*; *Holway et al., 2002*; *Tsutsui & Suarez, 2003*; *Lester & Gruber, 2016*). In fact, a recent host switch of local *Macrodinychus* parasitoids to invasive ants has been suggested to be responsible for the high parasitism rates found in *M. yonakuniensis* (15%) (*Breton, Takaku & Tsuji, 2006*), *M. multispinosus* (26%) (*Lachaud, Klompen & Pérez-Lachaud, 2016*) and *M. sellnicki* (up to 90%) (*González, Gómez & Mesa, 2004*; *Krantz, Gómez & González, 2007*).

## Possible counter-adaptations against macrodinychid mites

Hidden inside the pupal silk cocoons, the immature mites studied here are practically invisible to adult host workers. In contrast, once eclosed from the pupal cocoon, adult mites are exposed and thus are accessible for host inspection. Similar to socially integrated species such as the spider *Sicariomorpha maschwitzi* (*Witte et al., 2009*; *von Beeren, Hashim & Witte, 2012*) and the silverfish *Malayatelura ponerophila* (*Witte et al., 2009*; *von Beeren et al., 2011b*), adult *Macrodinychus* spp. were mostly ignored or unnoticed by host ants. Nonetheless, host workers regularly antennated adult parasitoids and ultimately attacked them, although at a relatively low level. Low levels of aggression towards myrmecophiles are still biologically relevant. For instance, soft bodied myrmecophiles such as the silverfish *M. ponerophila* were occasionally killed in behavioral assays (*Witte et al., 2008*; *von Beeren et al., 2011b*). We interpret the occasional attacks towards *Macrodinychus* mites as a behavior to fight off the adult parasitoid before host brood become infected with parasitoid larvae. However, *Macrodinychus* specimens survived these attacks unscathed owing to their protective morphology, embodied by a well-sclerotized cuticle and the possibility to retract all extremities into cuticular cavities (pedofossae) (see Figs. 2 and 3). A more efficient host defense might be the ants' behavior following the initial attacks. *Macrodinychus* spp. were often picked up by workers in laboratory nests and dumped at the ants' refuse site, outside the inner nest part where the parasitoid target, i.e., ant brood, is located. The adult mites were mobile and regularly re-entered the brood chambers in laboratory nests, only to be picked up and dumped at the refuse site again. In addition to this, the frequent emigrations of army ants might represent another counter-measurement to reduce a colony's total fitness loss imposed by parasites and parasitoids (*Witte et al., 2008*; *von Beeren et al., 2011a*) because parasites/parasitoids can be left behind at the abandoned nest site (*Witte, 2001*). Support for this hypothesis comes from an observation during a nest emigration of *Leptogenys distinguenda* initiated in the laboratory at the field site. We collected three *Macrodinychus* spp. adults at the abandoned nest site (a 1 m ×1 m ×1 m plastic box filled with leaf litter), in other words, the emigrated colony shed off these parasitoids.

## CONCLUSIONS

Parasitoidism and also parasitism of ants by mites is likely more common than hitherto known (*Campbell, Klompen & Crist, 2013*; *Lachaud, Klompen & Pérez-Lachaud, 2016*) and the cryptic lifestyles of mites inside ant nests has certainly hampered their discovery (*Skoracka et al., 2015*). In fact, the species studied here were chance finds that were initially overlooked. It is safe to say that many more macrodinychid mites await scientific discovery (see e.g., *Lachaud, Klompen & Pérez-Lachaud, 2016*) and we thus would like to encourage researchers to specifically screen ant brood for these fascinating and rather unexplored parasitoids.

## ACKNOWLEDGEMENTS

AB thanks the Acarology Summer Program 2016 at Ohio State University in Columbus, for the awesome class and in-depth discussions about mites and science. We thank the reviewers for their helpful comments, which improved the mansucript. The following people helped to collect samples from *L. distinguenda* colony emigrations: Deborah Schweinsfest, Magdalena Mair, Sebastian Pohl, Volker Witte, Daniel Schließmann, Max Kölbl, Hannah Kriesell, Stefan Huber, and Sofia Lizon à l'Allemand. We thank Marc Pfitzer who guided the imaging process, Andrea Hilpert who dissected part of the pupal cocoons, and Philipp Hönle for editing the video file. We thank Peter Schwendiger of the Museum of Natural History Geneva and Stefan Friedrich of the Bavarian State Collection of Zoology for a friendly process of specimen loans. Finally, we are particularly thankful to Volker Witte who sadly passed away in 2015. Without his dedication to the study of army ants and their associates, this study would not exist.

### Funding

Adrian Brückner was funded by a PhD scholarship from the German National Academic Foundation (Studienstiftung des deutschen Volkes) and by a George W. Wharton fellowship of the Ohio State University. This work was supported by two projects of the German Research Foundation (DFG: project numbers WI 2646/3 and BE 5177/4-1). Open access was supported by the German Research Foundation and the Open Access Publishing Fund of the "Technische Universität Darmstadt". The funders had no role in study design, data collection and analysis, decision to publish, or preparation of the manuscript.

### Grant Disclosures

The following grant information was disclosed by the authors:
German National Academic Foundation.
Ohio State University.
German Research Foundation: WI 2646/3, BE 5177/4-1.
Technische Universität Darmstadt.

## Competing Interests

The authors declare there are no competing interests.

## Author Contributions

- Adrian Brückner and Hans Klompen analyzed the data, contributed reagents/materials/analysis tools, wrote the paper, prepared figures and/or tables, reviewed drafts of the paper.
- Andrew Iain Bruce and Rosli Hashim performed the experiments, reviewed drafts of the paper.
- Christoph von Beeren conceived and designed the experiments, performed the experiments, analyzed the data, contributed reagents/materials/analysis tools, wrote the paper, prepared figures and/or tables, reviewed drafts of the paper.

## Data Availability

BOLD Systems; *Macrodinychus* {genus}:

http://www.boldsystems.org/index.php/Taxbrowser_Taxonpage?taxon=Macrodinychus.

## New Species Registration

The following information was supplied regarding the registration of a newly described species:

Publication LSID:

urn:lsid:zoobank.org:pub:84ADDB13-56F3-431D-9244-E19C3A2F7E04

Species names:

*hilpertae:*

urn:lsid:zoobank.org:act:88FADEC7-D4A5-4491-A45E-8F8176B65D31

*derbyensis:*

urn:lsid:zoobank.org:act:83C74026-EF1F-41A8-915F-49B0B2188E2A.

## Supplemental Information

Supplemental information for this article can be found online at http://dx.doi.org/10.7717/peerj.3870#supplemental-information.

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
