# Peer review of "Infection of army ant pupae by two new parasitoid mites (Mesostigmata: Uropodina)"

_PeerJ, doi:10.7717/peerj.3870_

## Round 0.1 · original submission · Minor Revisions

· Academic Editor

Minor Revisions

As you'll see, we were fortunate enough to obtain three careful reviews that indicated a variety of issues that need to be addressed before the manuscript can be accepted. However, given their nature, I'm confident that you'll be able to address them.

Reviewer 1 ·

Basic reporting

The writing was clear and easy to understand. The introduction provided adequate background information. There are some minor grammatical errors here and there that should be tended to.

Experimental design

• Line 98: If the pupae were collected from Aug 2008 to April 2011, then that spans far more than just 10 months. If the samples were collected from 10 specific months, please specific which months.
• Line 130: “Global parasitism rate” is too broad. I’d suggest the phrase “overall parasitism rate among Macrodinychus spp.”
• Line 132-133: It’s not clear why results on number of workers are assessed. There is not reporting on this data in the results section in terms of prevalence of infection.
• Given that the pupae were collected over a long span of time, perhaps one should look for potential seasonal or temporal patterns in infection rate.
• Line 441: Were the differences in prevalence of infection between colonies statistically different from one another?
• The behavior “antennated” is very similar to “ignored” except perhaps for the duration of antennation. Is this minor difference biologically relevant?
• Some of the methods require minor clarification. For instance, the experimental could have recorded the duration of the ant-mite interactions. Would this additional data not have been useful?

Validity of the findings

• I cannot comment specifically on the species descriptions, as this is not my area of expertise. I can comment only on the behavioral assays.
• Line 438-441: It’s not clear what value these set of numbers provide. The infection rate of 1.69% is already reported in line 434. What is the relevance of reporting number of workers?
• Lines 454-459: The reported values for AI are presumably in percentages, in which case the values seem exceedingly low to be biologically relevant. Can you please clarify?
• Line 452-453: Given that there are overall differences between the different myrmecophile species, it would be useful to present a post-hoc analyses to determine which groups are different from one another (as suggested in the methods).
• Line 474 and 488: Is there direct evidence that the mite is feeding on the hosts? The presence of a cavity does not necessarily indicate feeding. Even if the mites left behind a scar, this may be a response to the chelicerae pinching the host tissue, and not necessarily feeding by the mite. Still, please specify in the results section if scars were indeed observed in your study.
• Lines 483-484: This study does not actually provide experimental evidence that M. hilpertae and M. derbyensis feed on host hemolymph. This assumption is based on circumstantial evidence based on the presence of a cavity and scar, which could be a melanization response to the pinching/tearing of the chelicerae.
• Lines 490-509: The authors suggest that the overall low infection rate may be due to that fact that the association is native. But they failed to considered a number of other alternative explanations, e.g. seasonality, spatial (nest) heterogeneity, etc. The low infection rates may also be an artifact of sampling bias. Given that pupae were collected during colony migration, and that migration may serve as a form of counter-defense, then the pupae being transported may be biased towards uninfected pupae (assuming ants can detect the difference and leave behind the infected pupae).
• Figure 5A: The behavior “attack” is not included, is this an oversight?
• Figure 5B: Please indicate the units for the dependent variable.

Additional comments

• Abstract: The statement that “Our results support the hypothesis that the primary habitat of all Macrodinychus mites is the ant nest” is a somewhat of an overstatement and not really testable since one cannot conclude with full certainty that “all” Macrodinychus spp. inhabit nests. This statement is also inconsistent with lines 76-77, which state that only 1/3 of Macrodinychus species are associated with ants.
• Lines: 511-512: This statement seems out of place as there is not further reference to the down-regulation of parasitism rates and it’s relevance to the results.
• Lines 524-527: The rationale behind this statement is not clear. Please expand.
• Discussion: It would be useful if the authors could discuss the potential ecological and perhaps biocontrol ramifications of this very interesting form of symbiosis.
• Overall, the discovery and morphological description for the two new mite species is valuable. However, the results on rates of infection need to be more detailed and interpretation of the behavioral assays needs to be treated with more rigour; for instance, the discussion does not address the different behavioral responses of the ants. For instance, the discussion does not provide much context for the results reported in Figures 5A and 5B.

Annotated reviews are not available for download in order to protect the identity of reviewers who chose to remain anonymous.

·

Basic reporting

AS a reviewer I am making no claim for expertise in mite systematics, but enjoy a solid description of natural history and a fascination with the communities in less known places, and that is what the present manuscript contributes along with a description of two new mite species. The dense populations of eusocial insects provide unusual opportunities for parasites and certainly the conditions where previously undescribed forms may be found. Thus, I believe this paper has clear value and perhaps a dual audience of the few mite specialists out there combined with the more abundant students of ants.

I also found the paper generally well written, especially the introduction and methods (for which I have basically nothing to say). they are were very clear to someone tangential to the topic, but I think some organizational shifts in the results and discussion will make the content more accessible for those more interested in the parasitoid system than the necessarily very complex and detailed description of mite anatomy associated with species identification.

Overall, I think the changes I suggest should be straightforward. The main point is to be consistent in the order of presentation across sections. You open with life history and focus the discussion on life history (actually you forget to discuss the new species aspects there, largely because I think you used a results and discussion approach for describing M. hilpertae and M. derbyensis, but bury the behavior in the results below the species description. Easily fixed, I think.

1) Start the results with the Life History and nest observations (442-467) and follow them with your description of the two species, and end results with those descriptions
2) The genus description is not your result nor really is a key. Why not move this to the discussion as a section on Generic variation and keys to the species (my one area where I think rewriting beyond moving text is required) as you preface your key with the material at 187-217 prior to presenting 397-431.
3) Still keep 469-527 as the start of discussion, and the Conclusion stays at the end, below the key.
Smaller points: in the systematic note (194-198), you need to translate the German group terms first and then simply indicate for precision the term used by Hirchmann, or omit that presentation if this “Gangsystematik” simply hasn’t influenced the modern systematics of the group.
At 210, to what the “Diagnosis” applies is needed. I assume Macrodinychus, but in contrast to what other genera?
249-251: adjective first…. Female or male operculum, not operculum female.
270-281, you claim you make an estimate, but report a very precise number, and similarly at 452-467, make sure significant digits are relevant to the means reported, for example 23.60 +/- 10.46 really is just 24 +/- 10, but check throughout your paper for both issues that reoccur.
360: avoid relative terms, “Legs relatively short”, in a species description lacks information without a clear contrast.
376 where you say “2, rather than 3”; to what does the 3 refer?

Experimental design

The design came out of a chance discovery, and that's ok.

Validity of the findings

solid. Aren't new species by definition "novel."

Additional comments

Figures: Hey, just nice photos. When I read two mite species, one thinks really… you can be sure when your behavioral work often lumped the two together, and yet your photos show clearly two different animals, and they fall out separately in the key completing that story.

Perhaps most of my comments are all up in the basic reporting section, but that is the most relevant. Shouldn't take more than an afternoons work to make improvements to readability. The paper tells a good story, adds alpha taxonomy on an understudied group, and links the species to a well known host species.

·

Basic reporting

This is a nice and good written manuscript with some deficiency.

Experimental design

Acceptable.

Validity of the findings

The manuscript contains original and new spientific results.

Additional comments

This is a well-written and clear manuscript, with nice pictures and some nice drawings. The found species seems to be new, but the illustrations are not enough for a taxonomic paper. The digital pictures are nice, but not show exactly the small details of the investigated specimens. Need to add more drawings to the manuscript, the most important part of the description of the new species is the exact and clear illustration which can help the identification and recognisation of the species.

Kontschan in press paper is already published.

---

## Round 0.2 · accepted · Accept

· Academic Editor

Accept

Thanks for carefully addressing all of the comments by the reviewers. I believe that the manuscript is much stronger as a result.